# Analysis of the Three-Dimensional Structure of the Misocyclones Generating Waterspouts Observed by Phased Array Weather Radar: Case Study on 15 May 2017 in Okinawa Prefecture, Japan

Ryusho Imai [1,†] and Nobuhiro Takahashi [2,*]

1 Graduate School of Environmental Studies, Nagoya University, Nagoya 464-8601, Japan
2 Institute for Space-Earth Environmental Research, Nagoya University, Nagoya 464-8601, Japan
* Correspondence: ntaka@nagoya-u.jp; Tel.: +81-52-789-3492
† Current affiliation: Japan Meteorological Agency, Tokyo 105-8431, Japan.

**Abstract:** Tornadoes are one of the most severe meteorological phenomena on the earth and their high winds cause serious damage to society. It is well known that vortices (mesocyclone or misocyclone, depending on their scale) in convective clouds contribute to tornadogenesis. High temporal resolution radar observations are necessary to elucidate the mechanism of tornadogenesis because convective clouds change drastically over time. This study focused on waterspouts that occurred on 15 May 2017 near Okinawa, Japan. Using Phased Array Weather Radar (PAWR) data, which provide three-dimensional data with high temporal resolution (every 30 s), radar reflectivity factors and Doppler velocity data were used to detect the three-dimensional structure of vortices. Using PAWR data, vorticities and diameters of these misocyclones were detected every 30 s and their potential vorticities, which are only possible because of the three-dimensional observation by PAWR, were calculated to understand the vortex generation and advection. The structures of two misocyclones (MC1 and MC2) were detected from Doppler velocity patterns. Combined with the radar reflectivity analysis, MC2 can be divided into two misocyclones (MC2a and MC2b). Potential vorticity of MC1 increased with time, probably because an outflow from the strong echo enhanced the lower horizontal shear. Potential vorticities in MC2a and MC2b were conserved in each period, with MC2b being nearly twice as large as MC2a.

**Keywords:** waterspout; phased array radar; misocyclone

## 1. Introduction

A tornado is "a rapidly rotating column of air extending vertically from the surface to the base of a cumuliform cloud, often with near-surface circulating debris/dust when over land or spray when over water. Although its presence is not required, a funnel cloud is often visible and may partly or fully extend from the cloud base to the ground" [1]. Due to its violent and destructive circulation, the tornado is one of the most severe phenomena in meteorology. The occurrence of tornadoes is not limited to over land, but also over water which are called waterspouts. Waterspouts are relatively weak tornadoes compared to tornadoes over land.

Many tornadoes are known to be associated with cumulonimbus clouds (parent clouds) that have vertical vortex structures [2]. Fujita [3] first defined the horizontal scales of the vertical vortex in a cumulonimbus and applied them to the classification of the vortex; a vertical vortex is classified according to its horizontal scale such as a mesocyclone for mesoscale (4–400 km) vortex and a misocyclone for misoscale (0.04–4 km) vortex. In this study, a mesocyclone has meso-$\beta$ scale (4–40 km) and a misocyclone has miso-$\alpha$ scale (0.4–4 km), based on the definition by [3].

Tornadoes are generally classified into the following two types according to the difference in the generation process of cumulonimbus clouds [2]. One is the supercell type; a single, huge convective cloud that lasts a long time because the updraft and downdraft regions in the cloud are separated, often having mesocyclones inside and producing severe weather conditions such as downbursts and hails. Tornadoes occur inside or at the edge of the mesocyclone in the supercell. The other is the non-supercell tornado (NST), which occurs in non-supercell clouds; NSTs are often short-lived tornadoes with smaller horizontal scales and weaker wind speeds than supercell tornadoes. The waterspouts treated in this study are also classified as NSTs. NSTs, including waterspouts, are difficult to observe and predict because they develop and dissipate in a short period of time from relatively small convective clouds. So-called "storm chasers" mainly target supercells because it is more difficult to predict the occurrence of NSTs than supercell-type tornadoes [4].

Research to characterize the conditions and the processes that generate NSTs and to understand the life cycle of NSTs have been conducted through observations using aircraft, ships, photographs and radar, and numerical experiments. Golden [5,6] used aircraft photogrammetry and smoke tracers to classify the life cycle of waterspouts over Florida. He showed that waterspouts were generated by the stretching of vortices due to cumulonimbus updrafts, and that vorticity was supplied by shear lines generated by the gust front. Golden [6] showed that less than 5% of waterspouts occurred in isolated cumulus clouds. Simpson et al. [7] conducted observations and numerical experiments on the relationship between cumulus-scale in-cloud vortices and waterspouts in Florida. They showed that waterspouts tend to form near the outflow of a gust front caused by a shower because strong convergence at altitudes below the cloud base enhances the updraft and concentrates the vorticity region into smaller vortices. Wakimoto and Wilson [8] analyzed 27 NSTs that occurred in Colorado in the summer of 1987. They found that vortices were generated by shear instability on the convergence line, which coupled with updrafts inside the cumulus as it passed over the convergence line to generate the NSTs. Lee and Wilhelmson [9] conducted numerical experiments on the development of NSTs on local fronts. They showed that the gust-front-enhanced, horizontal shear increased vorticity and convergence at the gust front then generated updrafts and formed cumulonimbus clouds. This process caused stretching and contraction of the vortices and produced multiple NSTs. Fujiwara and Fujiyoshi [10] attempted to detect invisible waterspouts using Doppler lidar volume scans. They showed that vortices along shear lines produced by gust fronts were generated in the early stages of waterspouts.

Studies attempting to understand the structure of NSTs and their parent clouds have also been conducted through radar observations. Wakimoto and Lew [11] observed waterspouts in Florida during the CAPE observation campaign and reported the detailed radar reflectivity and Doppler velocity field of a typical parent cloud associated with a waterspout for the first time. They showed that waterspouts were generated from rapidly developing cumulus clouds. Sugawara and Kobayashi [12] analyzed a misocyclone structure generated on a shear line based on the radar observation over Tokyo Bay in 2007. In their case, the diameter of the misocyclone increased with decreasing altitude, suggesting the breakdown of a vortex. Sugawara and Kobayashi [13] analyzed the temporal changes in the vertical structure of multiple waterspouts that occurred in a narrow cold frontal rainband in 2006 with relatively coarse temporal resolution (8 min). Saito et al. [14] analyzed a merger of misocyclones, which are important in the tornado development process, observed in the TOMACS observation campaign area in 2011. The misocyclone had an interesting structure with an upward tilt of about 1 km to the east and a nearly constant diameter of 1.5 km. Because the resolution of the volume scan was 6 min, no further analysis was conducted. Van Den Broeke and Van Den Broeke [15] observed cumulonimbus clouds that generated waterspouts with a dual-polarization radar. They observed unusually high differential reflectivity (ZDR) values accompanied the storm and its initiating boundary. They partially explained these high values by a high density of dragonflies.

The conditions and processes of NSTs, including waterspouts, are generally well understood based on observations and numerical experiments. The structure and the life cycle of waterspouts have also been summarized through observations using aircraft and radar. However, detailed temporal changes in the internal structure of convective clouds that generate tornadoes have rarely been observed due to the lack of temporal resolution of radar. This is because conventional radar observations require more than a few minutes for a volume scan, making it difficult to understand the life cycle of vortex structures in convective clouds. Moreover, unlike tornadoes over land which can be verified after the damage occurred, witnessed information is essential for waterspouts; thus, there are few examples of observations of convective clouds that can generate waterspouts. Wurman [16] developed the Doppler on Wheels (DOW) which is used for tornado observation with very rapid scanning speed (e.g., more than eight rotations per minute). Even this type of radar requires about two minutes to complete the volume scan if it requires 15 elevation angles.

Phased Array Weather Radar (PAWR) [17], which has been developed in recent years, is one means of solving the problem of insufficient temporal resolution of radar; PAWR can observe the three-dimensional structure of precipitation clouds about every 30 s and can capture a rapidly changing convective cloud life cycle. In Japan, since the installation of PAWR in Osaka in 2012, PAWR observations have been deployed at six locations. Adachi et al. [18] observed the rapid development of misocyclones in a convective cloud with a vault structure in Osaka using PAWR. They utilized the high temporal resolution of PAWR to analyze the three-dimensional structure of the vortex and found that the upward stretching of the misocyclone caused increases in vorticity.

This study analyzed an internal structure of a convective cloud that generated waterspouts in Okinawa Prefecture, focusing on the three-dimensional structure of misocyclones, and related them to detailed temporal changes in the three-dimensional structure of the radar reflectivity of the convective cloud. The target of this study is the convective cloud that produced waterspouts off the coast of Yomitan Village, Okinawa, Japan, on 15 May 2017 (Figure 1). PAWR data were used to capture detailed temporal changes in the 3D structure of radar reflectivity and misocyclones.

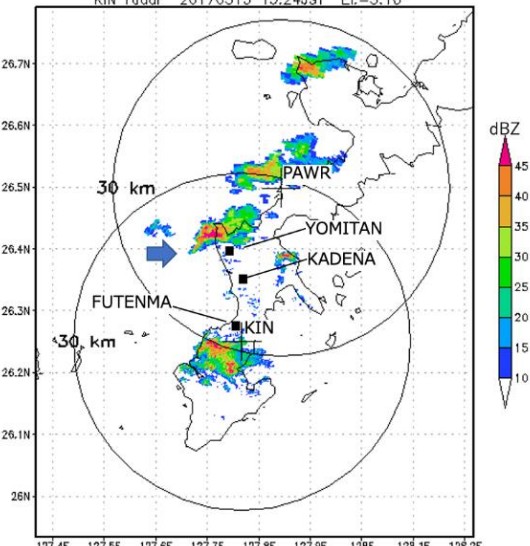

**Figure 1.** Locations of KIN radar and PAWR (+) with their 30 km observation range. Radar echo pattern (in dBZ) from KIN radar (elevation angle = 3.1°) at 15:24 JST (Japan Standard Time) on 15 May 2017. The target precipitation system is indicated by an arrow. The locations of Kadena Air Base, Marine Corps Air Station Futenma, and Yomitan village are indicated by squares.

Section 2 shows the data and analysis methods in this study. The analysis results are shown in Section 3. Section 4 discusses the formation mechanism and the life cycle of convective clouds and misocyclones. The conclusions are shown in Section 5.

## 2. Data and Analysis Methods

### 2.1. Data

For radar analysis, an X-band dual-polarization Doppler radar called KIN radar was used; it was owned by Nagoya University and installed at the Senbaru Campus at the University of the Ryukyus (Nishihara Town, Okinawa Prefecture) and Phased Array Weather Radar (PAWR) at National Institute of Information and Communications Technology (NICT) Okinawa Electromagnetic Wave Technology Center (Onna Village, Okinawa Prefecture). KIN radar operated at 3 rpm antenna rotation and observed 14 elevation angles in 6 min. Vertical polarization observations were not available during this observation period due to instrument problems. Therefore, only horizontal polarization data were used in this study. KIN radar has an observation radius of 60 km, and in this case, the distance to the target precipitation system was about 20 km. The PAWR is the same type of radar shown by [17], which can observe a 60 km range in 30 s with no elevation gap and can observe horizontal polarization radar reflectivity factor and Doppler velocity. Figure 1 shows the locations of KIN radar and PAWR and their 30 km ranges on a PPI map of reflectivity at an elevation angle of 3.1 degrees observed by KIN radar at 15:24 JST on 15 May 2017. The precipitation system analyzed in this study was the echo located between PAWR and KIN radar (indicated by an arrow in Figure 1).

Reports on the occurrence of tornadoes and waterspouts were issued as "Gust database of tornadoes, etc." by the Japan Meteorological Agency (in Japanese). According to this report, a waterspout was observed near the main island of Okinawa at 15:32 JST on 15 May 2017 but it was different from the target precipitation system in this study according to the Naha Local Meteorological Observatory (personal communication). According to aviation weather reports (METAR/SPECI reports) by Kadena Air Base and Marine Corps Air Station Futenma, waterspouts were observed at Kadena from 15:35 JST to 15:43 JST (WSK1) and from 15:50 JST to 15:54 JST (WSK2) (both locations unknown). The target precipitation system was the closest echo over the ocean to Kadena. At Futenma, a waterspout was observed at about 16 km northwest between 15:40 JST and 15:45 JST (WSF). Both results indicate that these waterspouts were generated in the precipitation system shown in Figure 1. It is conceivable that the waterspouts were generated from the precipitation system that produced the misocyclone but the correspondence between the two is not clear.

### 2.2. Radar Analysis

In this study, a misocyclone was detected from the Doppler velocity data, assuming a Rankine-type vortex. The Rankine vortex, a model that describes a rigid body rotation like a tornado, is used to calculate the tangential velocity of the vortex [2]. To detect misocyclones from PPI scan data of Doppler radar, the methods of [19,20] were applied as follows:

(1)    Find a pair of maximum and minimum values of Doppler velocity;
(2)    The angle between the line perpendicular to the line between the positions of the maximum and minimum of Doppler velocities in (1) and the radar beam direction should be 45° or less.

For the misocyclones detected by the above method, using the diameter $D$ from the Rankine vortex, the vertical vorticity ($\zeta$) can be approximated as follows:

$$\zeta \sim 2\frac{\Delta V_{tan}}{D},  \tag{1}$$

where, $\Delta V_{tan}$ is the difference in the tangential velocity of the vortex. The diameter of the vortex was defined as the distance between points of the maximum and minimum Doppler velocity and the difference in the tangential velocity was defined as the difference between

the maximum and minimum Doppler velocities. Because PAWR observations have no gap in elevation, the vortex can be identified continuously in the vertical direction.

Potential vorticity was used to evaluate the effects of generation and advection in the detected vortex. Potential vorticity is a dynamical quantity that is conserved when an air column with a certain vorticity between two isentropic planes moves adiabatically and frictionless. The potential vorticity is usually constant; however, it changes as new vorticity is generated or vorticity advection occurs. The potential vorticity (*PV*) on the horizontal isentropic plane is defined as:

$$PV = -g(f + \zeta)(\partial\theta/\partial p), \tag{2}$$

where, $f$ is the Coriolis parameter, $g$ is the gravitational acceleration, $\theta$ is the potential temperature, and $p$ is the atmospheric pressure. The value $-(\partial\theta/\partial p)$ means the stability of the atmosphere between two isentropic planes. For the vortices targeted in this study, the potential temperature gradient can be assumed to be constant due to the small vertical motion of the vortices and the short time scale. Assuming hydrostatic equilibrium, the potential vorticity PV is expressed by the ratio of the absolute vorticity $(f + \zeta)$ to the thickness $h$:

$$PV = \frac{f + \zeta}{h} \tag{3}$$

Since the Coriolis parameter $f$ is small relative to the vertical vorticity $\zeta$, the potential vorticity *PV* can be approximated to the ratio of the vertical vorticity $\zeta$ to the vortex thickness $h$. The Coriolis parameter $f$ at the latitude 26°N (the latitude of Okinawa) is about three orders of magnitude smaller than the vorticity ($10^{-2}$ s$^{-1}$) described below. Therefore, it is appropriate to approximate the potential vorticity PV as the ratio of the vertical vorticity $\zeta$ to the vortex thickness $h$.

In this study, the potential vorticities were calculated by averaging the calculated vortices for each elevation angle in the vortex tube (continuously detected vortices in the vertical direction). The depth of the vortex tube was defined as the difference in altitude between the maximum and minimum elevation angles of the vortex tube. If a vortex was detected at only one elevation angle, the potential vorticity was not calculated. If the vertical distance of 0.5 km or more between the two detected vortices, they were treated as vertically separated vortices.

In this study, we obtained low-level (≤2 km) horizontal wind fields by dual Doppler analysis using KIN radar and PAWR. The resolution of the data is 0.2 km horizontally and 0.5 km vertically, with a time interval of 6 min to match the operation cycle of KIN radar.

## 3. Results

### 3.1. Environmental Condition

According to the surface weather chart at 09 JST (Japan Standard Time = UTC + 09:00) and 15 JST on 15 May 2017, a Baiu front (a stationary precipitation zone that appears in June and July near the Japanese Islands) was stagnant near the main island of Okinawa; a precipitation system associated with the Baiu front was forming over the region as shown in Figure 2 which shows surface weather chart at 09 JST. According to [21], the Baiu frontal zone is characterized by "a narrow steady precipitation zone, strong gradient of equivalent potential temperature, thick moist neutral layer, steady generation of convective instability". It is also characterized by a weak convergence zone with a southerly component of winds prevailing south of the front and a northerly component of winds prevailing north of the front. In this case, the characteristics of the Baiu front were also evident, according to reanalysis data from ERA5 [22], where the precipitation system corresponded to a large gradient of water vapor in the lower levels (<850 hPa), with relatively weak wind speeds in the lower levels, shifting to southeast winds up to about 300 m altitude and southwest winds above 300 m. The magnitude of the vertical shear between the surface and at an altitude of 800 hPa was 3.7 m s$^{-1}$.

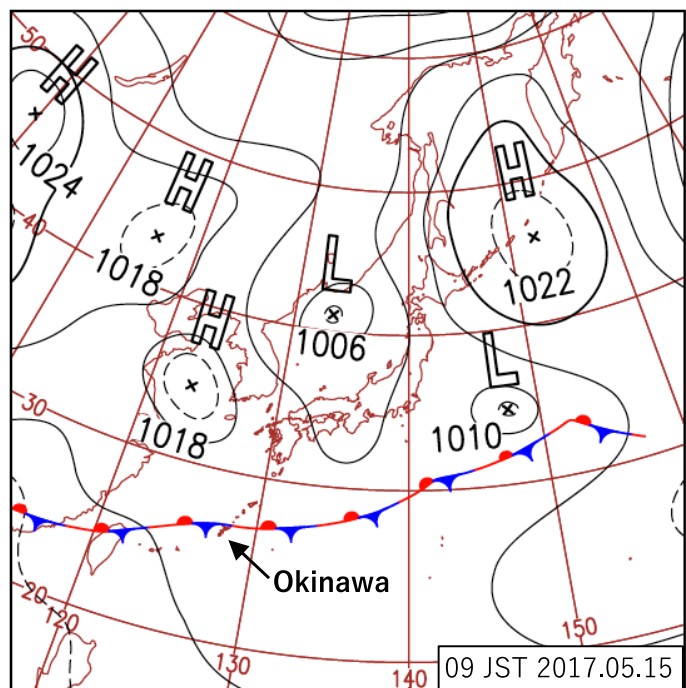

**Figure 2.** Surface weather chart from Japan Meteorological Agency at 09 JST on 15 May 2017.

In general, Convective Available Potential Energy (CAPE) and Storm Relative Helicity (SRH) are used as indices of the environmental conditions for tornado occurrence. Typical values of the environmental field for tornadoes associated with supercells over the North American Great Plains are CAPE > 1500 J kg$^{-1}$ [2] and SRH$_{3km}$ > 150 m$^2$ s$^{-2}$ (SRH between the ground and 3 km in height) [23]. In particular, a large SRH$_{1km}$ (SRH between the ground and 1 km in height) is an indicator of a mesocyclone relatively close to the surface, which is an indicator of a tornado in a supercell [24,25]. The environment for tornado occurrence in Japan differs from that in North America in that the middle troposphere is often warm and humid and relatively small CAPE = 1185 J kg$^{-1}$ has been reported as the atmospheric environmental condition during tornado occurrence [26].

Atmospheric environment parameters were calculated for three sounding stations surrounding the main island of Okinawa (Naze, Ishigakijima, and Minamidaitojima) on 15 May 2017. The maximum values were as follows: CAPE = 131.25 J kg$^{-1}$, SRH$_{3km}$ = 72.51 m$^2$ s$^{-2}$, and SRH$_{1km}$ = 27.56 m$^2$ s$^{-2}$, indicating that both CAPE and SRH were much smaller than the atmospheric conditions of the supercell tornado. The lifted condensation level (LCL) at these three stations at 09 and 21 JST were 1600 m (09 JST) and 1330 m (21 JST) at Naze, 860 m (09 JST) and 620 m (21 JST) at Ishigakijima, and 950 m (09 JST) and 700 m (21 JST) at Minamidaitojima. The cloud base heights at these three sites based on ERA5 were 1320 m (Naze), 401 m (Ishigakijima), and 442 m (Minamidaitojima), which were similar to the LCL at 21 JST at Naze, but 200 to 400 m lower than the LCL at the other sites. The value of the area where the target echoes occurred was even lower at approximately 330 m from ERA5. The METAR from Kadena reported the cloud height was about 580 m

### 3.2. Analysis of Radar Echo

Figure 3 shows the radar reflectivity factor field at 6 min intervals from 15:12 JST (hereafter JST is used as the time) to 15:48 obtained from KIN radar PPI observations (elevation angle 3.1 degrees). This figure shows the target precipitation system from the early development stage to the dissipation stage. In Figure 3, the echo began to be observed at around 15:12. Subsequently, the radar reflectivity factor increased as the echo propagated eastward with an average speed of 2.8 m s$^{-1}$ at the west end of the echo and 5.6 m s$^{-1}$ at the east end of the echo; the echo area expanded horizontally as it developed vertically.

The radar reflectivity factor peaked around 15:30, with the maximum radar reflectivity factor exceeding 50 dBZ. A linear echo with a northeast–southwest direction formed on the western edge of the echo from 15:24 to 15:30. In the Doppler velocity field in Figure 3, a pair of Doppler velocity maximum and minimum (MC1 in Figure 3) began to be seen near the linear echo at the western edge at about 15:30 and two pairs (MC1 and MC2) were identified at 15:36. In this figure, MC2 was detected until about 15:42. PAWR detected the same vortices from 15:26 to 15:47. In Figure 3, KIN radar observed these vortices from the south–southeast, so the pairs of the maximum and minimum Doppler velocities were aligned roughly east–west. Figure 4 shows a three-dimensional structure (side view from southeast and top view from southwest) observed by PAWR at 15:30. At this time, the echo had almost reached its maximum altitude (5 km) at the high reflectivity region in the northeastern part of the echo. On the other hand, at the western edge where the vortices were observed, the echo top was found to be as low as 3 km in height, and the strong echo region remained below 1 km in height. The strong echo in the northeastern part of the echo had weakened afterward. Between 15:30 and 15:36, the shape of the strong echo region changed from a northeast–southwest alignment to an east–west direction. In particular, the echo at the west end corresponding to the vortex shows that a new strong echo has formed to the north of the existing echo, which contributed to the change in the shape of the echo.

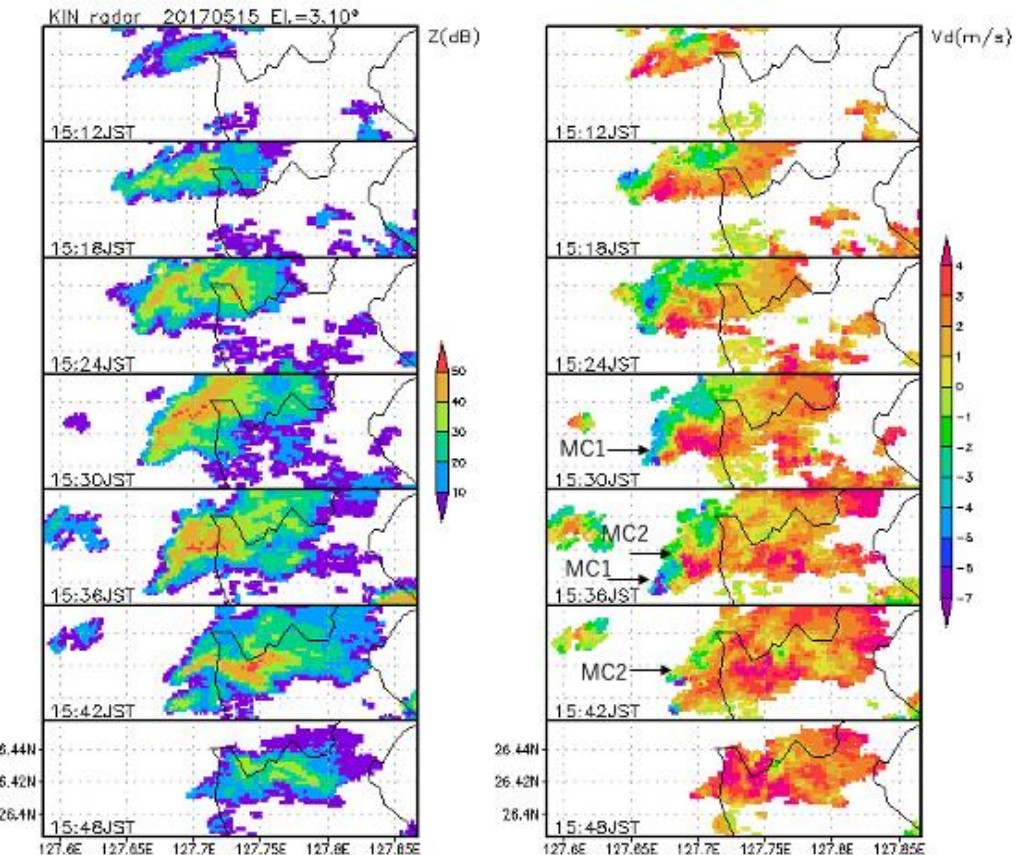

**Figure 3.** Radar reflectivity field (**left**) and Doppler velocity field (**right**) from KIN radar (elevation angle = 3.1°) from 15:12 to 15:48 JST. The local maximum/minimum pairs of Doppler velocity which correspond to two misocyclones (MC1 and MC2) analyzed in this study are indicated by arrows.

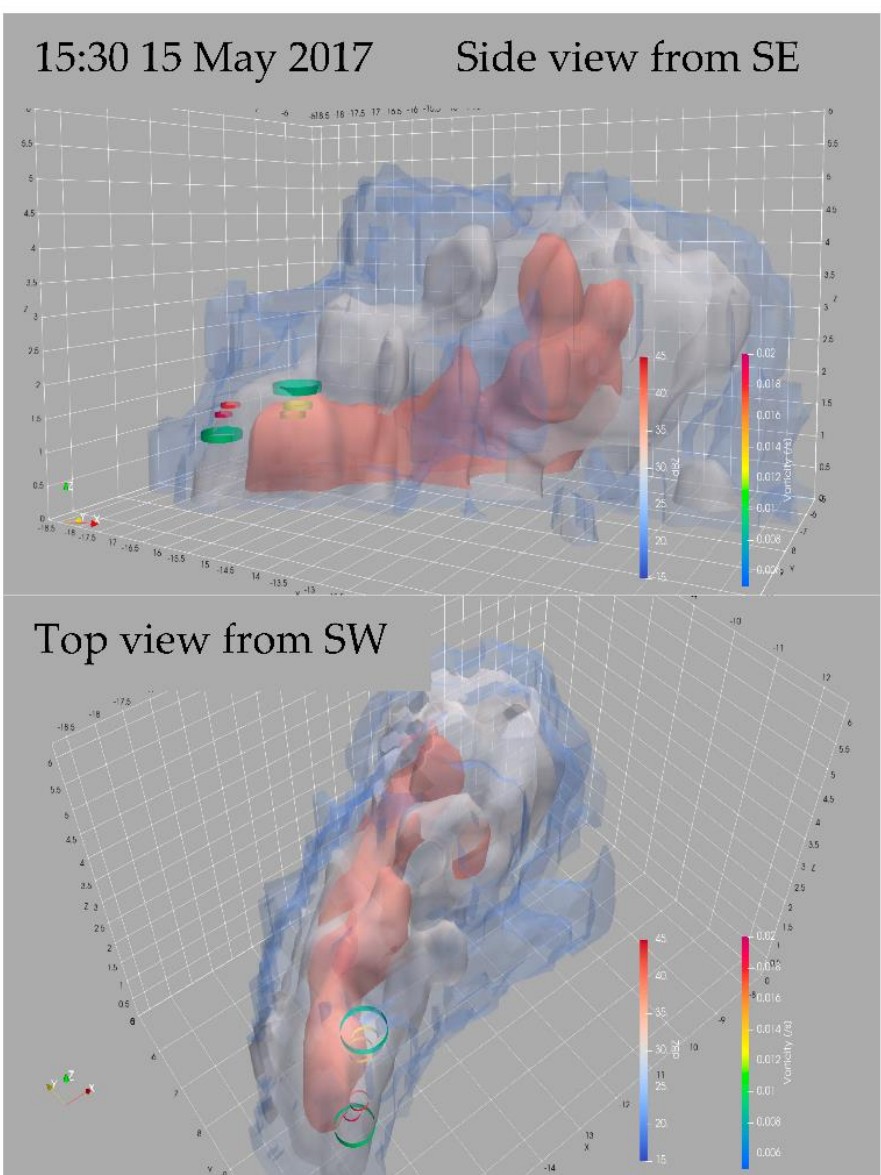

**Figure 4.** Three-dimensional view: ((**top**) side view from southeast, (**bottom**) top view from southwest) of the radar echo from PAWR at 15:30 JST on 15 May 2017. Circles in this figure show the location, size, and vorticity of the misocyclones estimated from the Doppler velocity data of PAWR.

### 3.3. Dual Doppler Analysis

Figure 5 shows the horizontal wind fields at altitudes of 0.5 km, 1.0 km, 1.5 km, and 2.0 km at 15:24 and 15:30 superimposed on the radar reflectivity. At 15:24, easterly winds were prevailing at an altitude of 0.5 km in this precipitation system. At an altitude of 1.0 km, southeasterly winds were converging in the high reflectivity region from the south of the echo. Additionally, northerly winds were present at the western end of the echo, indicating that the precipitation system had a cyclonic circulation. At an altitude of 1.5 km, the southerly and westerly winds converged on the eastern part of the echo and southeasterly and northwesterly winds converged at the western part of the echo. At an altitude of 2.0 km, the westerly winds were dominant over the entire echo. At 15:30, at altitudes of 0.5 km and 1.0 km, the strong echo region was dominated by weak easterly or northeasterly winds, and the southeasterly winds entered from the south side of the strong echo region, forming a convergence. At altitudes of 1.5 km and 2.0 km, westerly or northwesterly winds were seen in the intense echo region, forming a convergence on the

southern part of the intense echo region. During this period, the northerly component was observed at the western edge of the echo up to an altitude of 1.5 km.

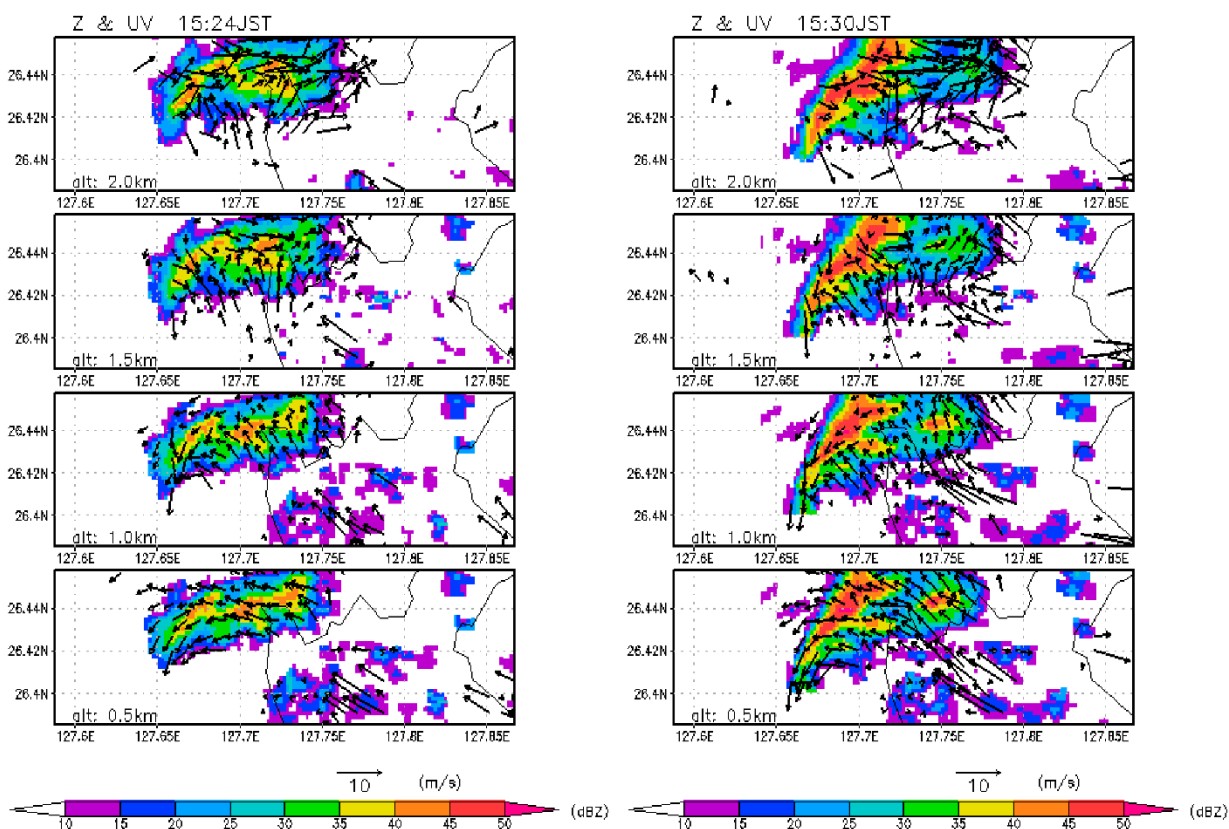

**Figure 5.** Horizontal wind field overlayed on the radar echo at altitudes of 0.5, 1.0, 1.5 and 2.0 km at 15:24 JST and 15:30 JST obtained from dual Doppler analysis using KIN radar and PAWR.

### 3.4. Misocyclones

As shown in Figure 3, two vortices (maximum and minimum Doppler velocity pairs) were observed from 15:30 to 15:42 based on the KIN radar observation. These vortices were classified as misocyclones according to the classification of [3]. The first observed vortex is called MC1 and the other is called MC2 (Figure 3). From the Doppler velocity pattern of PAWR, these vortices were detected almost continuously in three-dimensions; MC1 was detected from 15:25:30 to 15:36:30 and MC2 was detected from 15:28:30 to 15:47:00.

#### 3.4.1. Time Variation of Diameters and Vorticities of MC1 and MC2

Figure 6 shows the time-height cross-sections of the misocyclone diameter and vorticity every 30 s. MC1 (Figure 6, top) was detected at 15:25:30 (time is showed in 30-second steps according to the observation cycle of PAWR) at an altitude of 2.1 km with a diameter of about 1.7 km and a vorticity of about $1.1 \times 10^{-2} \text{ s}^{-1}$, then expanded to a lower altitude (0.7 km) maintaining its diameter and vorticity. At 15:30:30, the vortex changed to a smaller diameter and higher vorticity; the diameter was about 0.5 km and the vorticity was about $2.9 \times 10^{-2} \text{ s}^{-1}$. At 15:31:00 It reached the lowest detectable altitude (the lowest echo height) of about 0.4 km, with a diameter of about 0.6 km and a vorticity of about $2.3 \times 10^{-2} \text{ s}^{-1}$. After 15:32:00, the low-level misocyclone disappeared, the misocyclone existed at an altitude of 1.1 to 1.6 km, and the vorticity decreased to about $2.0 \times 10^{-2} \text{ s}^{-1}$ maintaining its diameter. After 15:36:30, MC1 was no longer detected.

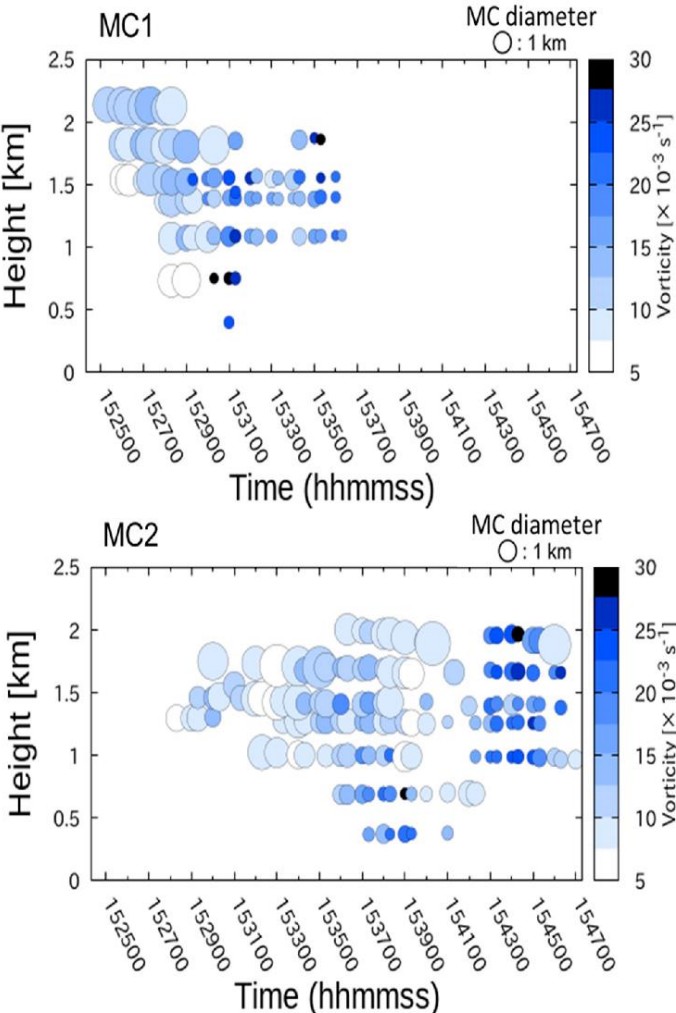

**Figure 6.** Time-height series of misocyclones: ((**top**) MC1, (**bottom**) MC2). Size of the circles expresses the diameter of the misocyclones and the color of the circles expresses the vorticity.

The MC2 (Figure 6, bottom) was detected at an altitude of 1.3 km at 15:28:30 with a diameter of about 1.3 km and a vorticity of about $0.9 \times 10^{-2}\,\text{s}^{-1}$. At 15:32:00, the vortex expanded about 0.3–0.5 km downward. At 15:36, it expanded to an altitude of 0.7–2.0 km with a diameter of 0.9 km and a vorticity of $0.9 \times 10^{-2}\,\text{s}^{-1}$ at an altitude of 0.7 km. At 15:37:30, the lowest altitude of MC2 reached its lowest detectable altitude of about 0.4 km, with a diameter of about 0.7 km and a vorticity of about $1.5 \times 10^{-2}\,\text{s}^{-1}$. At 15:39:00, the diameter was about 0.6 km and the vorticity was about $2.8 \times 10^{-2}\,\text{s}^{-1}$ at an altitude of 0.7 km. MC2 maintained its characteristics until 15:40:30. Subsequently, vortices were detected intermittently in space and time until 15:42:30. After 15:43:00, MC2 was detected at an altitude from 1.0 to −2.0 km, a diameter of about 0.7 km and a vorticity of about $2.0 \times 10^{-2}\,\text{s}^{-1}$, which maintained until 15:47:00.

### 3.4.2. Temporal Change of 3D Structure of Radar Echo and Misocyclones

In order to see the relationship between the development of radar echo and changes in the diameter, the vorticity, and the length of MC1 and MC2, the relative positions of those vortices to the echo were analyzed three-dimensionally (Figures 7 and 8). Figure 7 shows a three-dimensional image of MC1 viewed from the southeast at 2-minute intervals starting at 15:27. In this figure, MC1 is represented by circles, the diameter of MC1 is represented by the diameter of the circle, and the vorticity is represented by the color. The vortex was located south of the strong echo region, as shown in Figure 4, maintaining its relative position to the echo. At 15:29 and 15:31, MC1 tilted to the northeast with altitude (or rather,

the lower level shifted to the southwest). This may be due to the expansion of the strong echo area in the lower levels. From 15:29 to 15:31, the vortex diameter decreased and the vorticity increased, as described in Figure 6, and the echo gradually weakened after 15:29. In particular, the strong echo area above 2 km altitude weakened at 15:31 and at 15:33 a new strong echo area was formed on the west side, east of the MC2 formation location. Figure 8 shows the same as Figure 7 but for MC2; MC2 was located south of the strong echo area as was MC1 but the echoes corresponding to MC2 were different at 15:38 and 15:43. The temporal variation of MC2 from the three-dimensional radar echoes indicates that the vortex was continuously detected at almost the same location from the Doppler velocity data but was accompanied by different echoes in the first and the second half of the MC2 duration (corresponding to a discontinuity at around 15:41 in Figure 6). The echo corresponding to the first half of MC2 (called MC2a) weakened around 15:40, followed by a newly formed weak echo at 15:43, which corresponded to the second half of MC2 (called MC2b). MC2b was tilted to the east with altitude along the echo envelope.

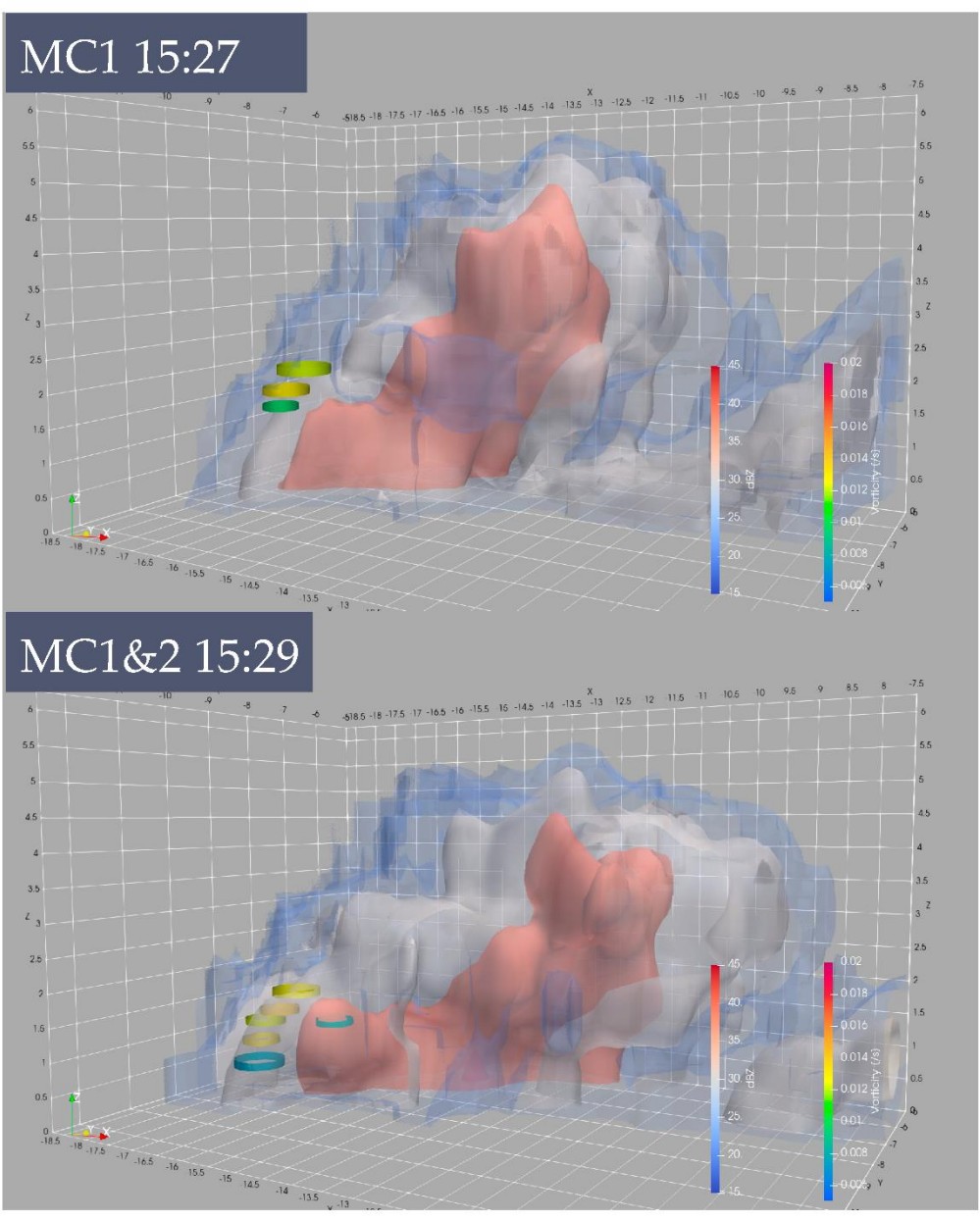

**Figure 7.** *Cont.*

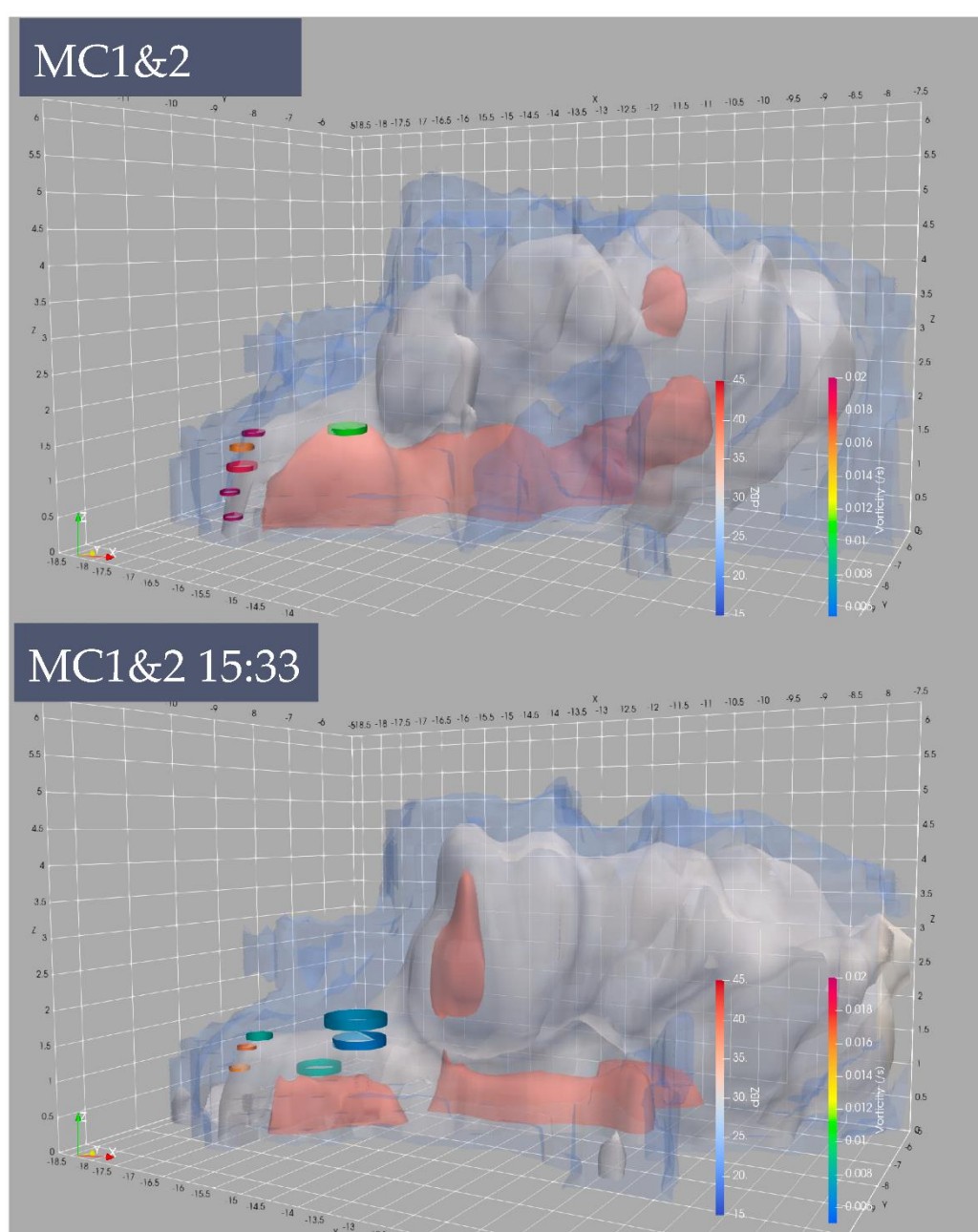

**Figure 7.** Same as Figure 4 except for side view from southeast at 15:27, 15:29, 15:31 and 15:33 JST.

### 3.4.3. Potential Vorticities of MC1 and MC2

Considering that the misocyclone generated at least 400 m in height and relatively weak echo continued at the misocyclone, we may apply the adiabatic and frictionless condition to assume the potential vorticity is conserved in the absence of vorticity generation or advection of these misocyclones. The potential vorticity is used to evaluate the development process of misocyclones. Figure 9 shows the time series of potential vorticities and vortex tube lengths for MC1 and MC2. The potential vorticities of MC1 showed a zigzag pattern that correlated well with the length of the vortex tube with time, with the potential vorticity ranging from 0.62 to $2.8 \times 10^{-2}$ km$^{-1}$ s$^{-1}$. This indicates that the estimated potential vorticities may be affected by the detectability of the vortices by PAWR. Figure 8 shows that the vortex tube length clearly expanded after 15:28 and the potential vorticity decreased correspondingly. Figure 5 shows that vortex characteristics (diameter and vorticity) changed significantly after 15:30 in MC1, while the potential vorticity increased after 15:30 according to Figure 9.

The time series of potential vorticity in MC2 (Figure 9b) is divided into two parts; from 15:33:00 to 15:40:30, the potential vorticity of the vortex tube (MC2a) was almost constant, except around 15:35:00, at about $0.7 \times 10^{-2}$ km$^{-1}$ s$^{-1}$ with a large change in vortex tube length,; from 15:42:00 to 15:45:30, the potential vorticity of the vortex tube (MC2b) was almost constant at about $1.7 \times 10^{-2}$ km$^{-1}$ s$^{-1}$. The separation of the two parts corresponded to the separation of the misocyclones (MC2a and MC2b) noted in Figure 8.

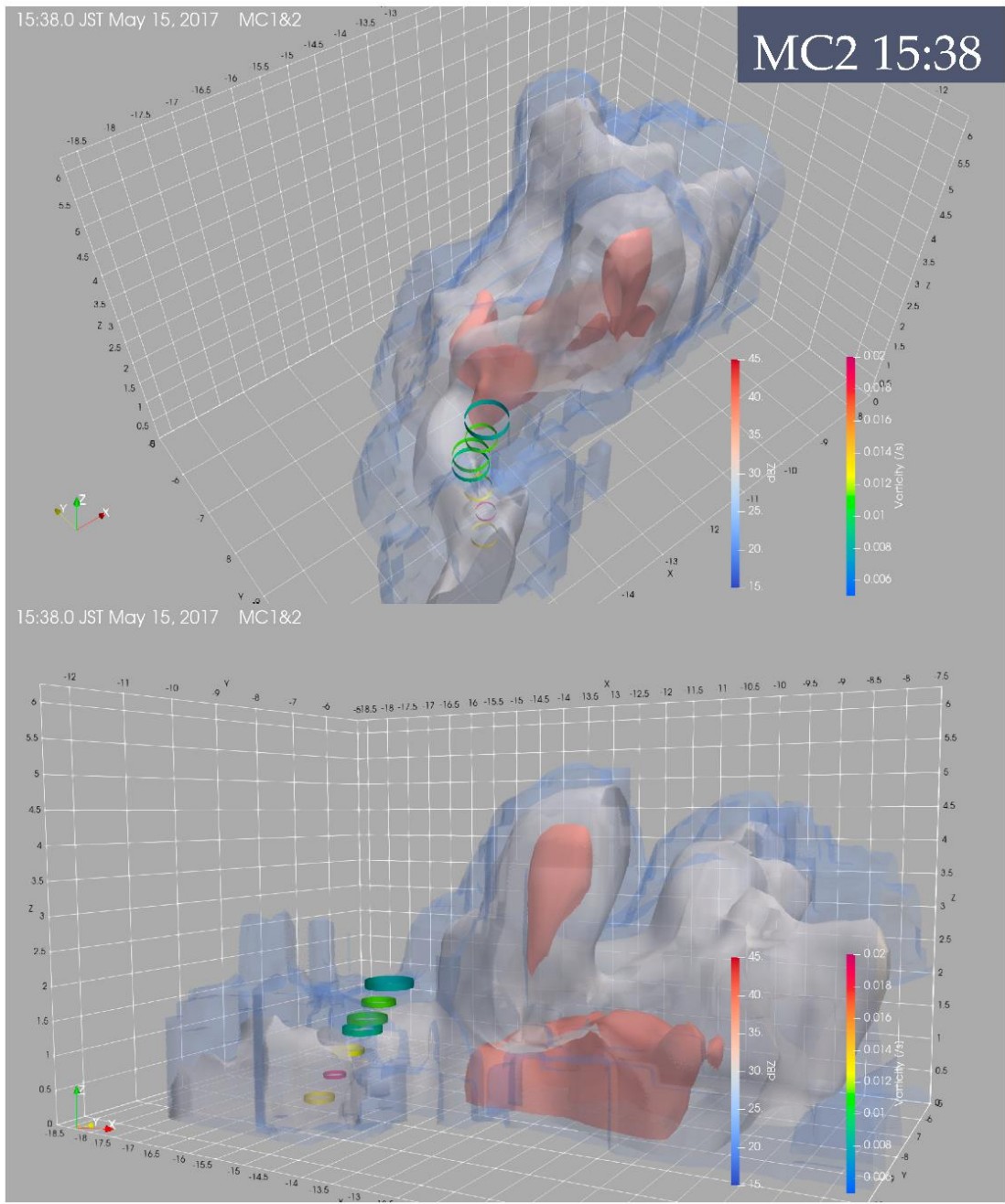

**Figure 8.** *Cont.*

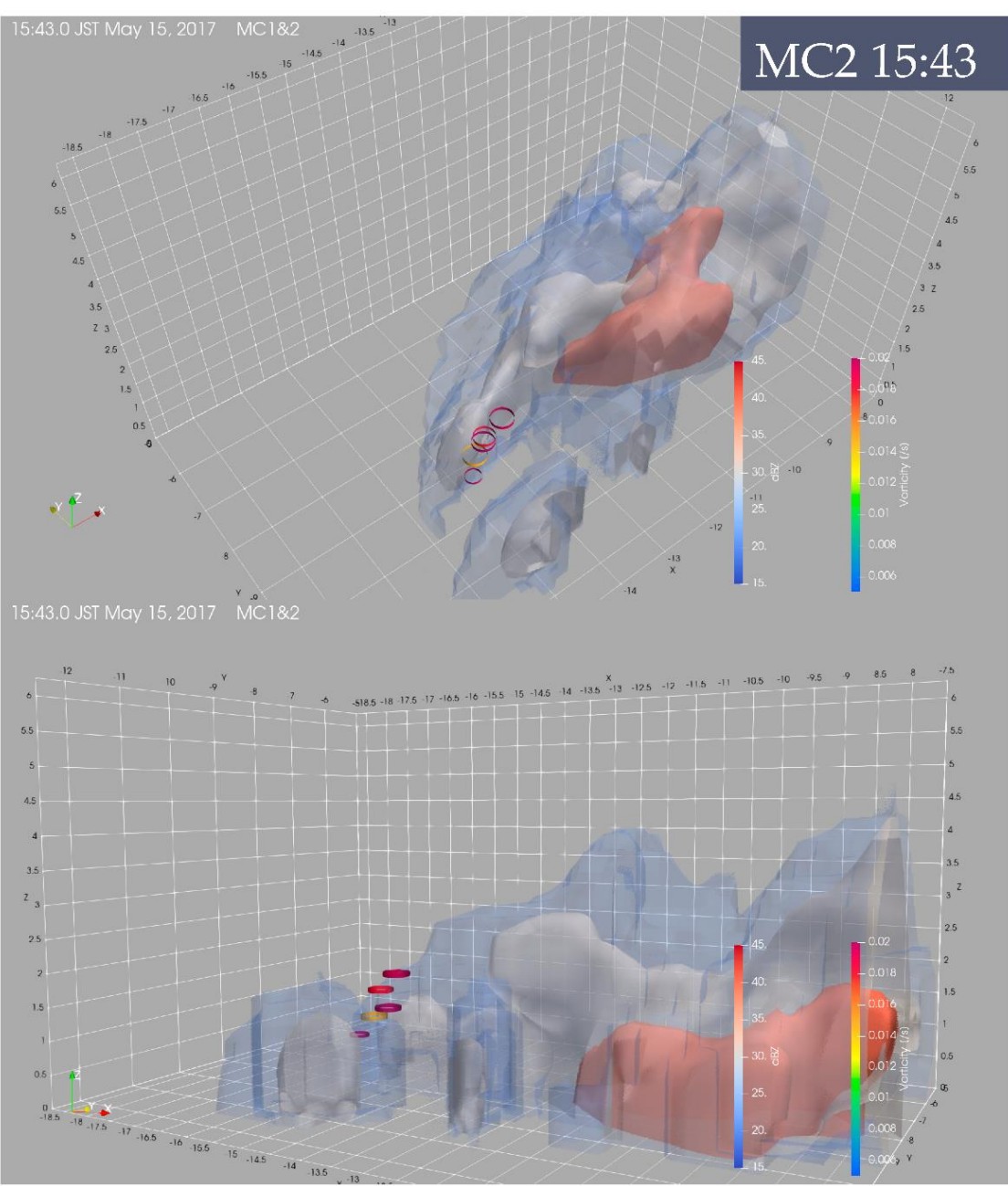

**Figure 8.** Same as Figure 4 except for 15:38 JST and 15:43 JST.

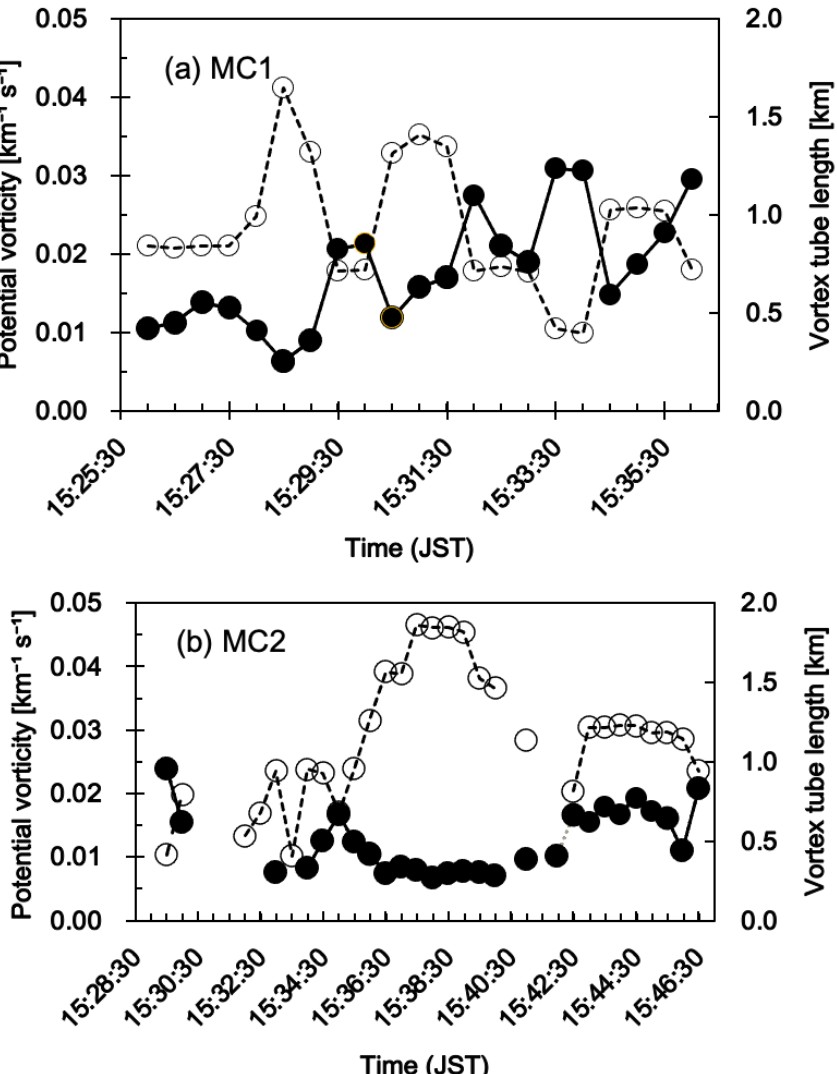

**Figure 9.** Time series of potential vorticity (solid circles) and length of the vortex tube (open circles) of (**a**) MC1 and (**b**) MC2.

## 4. Discussion

First, we discuss the environmental conditions and the life cycle of the convective cloud associated with the formation of the misocyclone. Its precipitation system formed south of the disappearing preceding precipitation system (located north of the target echo in Figure 1). Dual Doppler analysis confirmed the presence of horizontal shear due to northerly winds and environmental easterly winds below 2 km altitude. The northerly winds are thought to be an outflow from the preceding precipitation system. In the ERA5 wind data, a northerly component appears below 700 hPa north of the Baiu front. This suggests that the northerly component was somehow transported by the preceding northern precipitation system. It may be similar to the case of waterspouts analyzed by [7] which occurred in convergence zones formed by outflows from precipitation showers.

The formation situation of the misocyclone was different between MC1 and MC2. MC1 was found at the mature stage of the precipitation system, which developed to its maximum altitude around 15:27 (Figure 7). The vorticity and the vortex diameter of MC1 changed significantly around 15:30. This change coincided with the timing of the vortex tilt shown in Figure 7. This vortex tilt was probably due to the strengthening of the divergent flow in the lower layers as the strong echo weakened, corresponding to the enhancement of

northeasterly winds below 0.5 km in Figure 3. It can be inferred that these winds increased the horizontal shear, leading to an increase in potential vorticities.

As shown in Figures 8 and 9, MC2 can be divided into two vortices, MC2a and MC2b, based on the 3D radar echo pattern. Assuming that the potential vorticity was conserved in each mesocyclone, the discontinuity of the potential vorticity during MC2 can be explained. The formation of MC2a corresponded to the formation of the new strong radar echo seen at 15:33 in Figure 7, with horizontal shear still existing. On the other hand, MC2b also appeared, forming another new echo though this was not as well-developed. This does not explain why the potential vorticity of MC2b was larger than that of MC2a in relation to the development of echo. Potential vorticities in MC2a and MC2b were nearly constant in each period. In particular, the length of the vortex tube changed with time in MC2a, suggesting that the updraft associated with the development of the echo changed the length of the vortex tube but new vorticity was not supplied; in MC2b, the vortex tube length remained almost constant and the echo was weak, suggesting that no new vorticity generation or advection occurred. The condition for the potential vorticity to be constant could be that the initially given vortex is conserved, or it could be that the vortex supply was balanced by the consumption of the vortex due to friction; further research is needed to determine this condition. In addition, this study tried to apply the potential vorticity to the characterization of misocyclones assuming the hydrostatic equilibrium and shallow water approximation for such severe phenomena. Further discussion is needed on this point. This paper just demonstrated the possibility to estimate potential vorticity if PAWR is available.

Regarding the relationship between misocyclones and waterspouts, MC1 was detected from 15:25:30 to 15:36:30, MC2a from 15:32:00 to 15:40:00, and MC2b from 15:42:00 to 15:47:00. Waterspouts were observed from 15:35 to 15:43 (WSK1), from 15:40 to 15:45 (WSF), and from 15:50 to 15:54 (WSK2) according to the METAR/SPECI reports. MC2a and MC2b roughly coincided with the times reported by WSK1 and WSF, respectively; however, the exact locations of the waterspouts are unknown, so no further correspondences can be examined.

The advantages of using PAWR for the detection of misocyclones are discussed: observation of a three-dimensional structure without gap and fast electrical scan to complete three-dimensional observation in 30-second. The former has the advantage of obtaining the vertical structure of misocyclones without gaps, although PAWR cannot detect vortices smaller than the radar resolution or those not accompanied by precipitation. This also makes it possible to estimate the potential vorticities, which is useful for estimating the mechanism of vortex formation. For the latter, MC2 was divided into two vortices by high temporal resolution observations, it is suggested that a temporal resolution of about 30 s is necessary.

## 5. Conclusions

Tornadoes are known to be generated from rapidly developing convective clouds; mesoscale or smaller-scale circulations in clouds (mesocyclones or misocyclones) contribute significantly to tornado development. With its high temporal resolution and three-dimensional observation capability, PAWR is a powerful tool for tracking the three-dimensional structure of mesocyclones/misocyclones and the development process of convective clouds (parent clouds). In this study, PAWR was used to describe the three-dimensional structure of misocyclones (vorticity, diameter, and height) in the convective cloud that produced waterspouts. The purpose of this study is to analyze the detailed temporal variation of misocyclones, along with the three-dimensional structure of radar reflectivity, and to characterize misocyclones. The analysis was based on case studies of waterspouts observed off the coast of Yomitan Village, Okinawa, Japan, on 15 May 2017. Environmental conditions for this case were examined using atmospheric sounding data. CAPE and SRH, which are indicators of convective activity and tornado generation, were much smaller than typical values for supercell tornadoes in Japan. Weak vertical shear was

observed around 1.0 to 1.5 km altitude from the sounding data. Dual Doppler analysis of PAWR and KIN radar was used to estimate the low-level horizontal wind field. Northerly winds were analyzed at the western edge of the precipitation system where misocyclones were detected, which were probably generated by the preceding precipitation system. This horizontal shear formed by the northerly and environmental easterly to southerly winds is thought to have generated the misocyclone.

Two misocyclones, MC1 and MC2, were detected at the western edge of the echo based on the Doppler velocity pattern. These misocyclones formed just south of the strong echo region and MC1 showed a large change in potential vorticity with time. This could be due to the detection of vortex tubes by radar since the potential vorticity and the vortex tube length correlated well. Even without this, the potential vorticity appeared to increase with time. This is explained by the enhancement of low-level horizontal shear by the outflow from the strong echo. MC2 was divided into two vortices, MC2a and MC2b, which could be determined from the 3D structure of the echo with a very high temporal resolution (30 s). This was also consistent with the fact that the potential vorticity was discontinuous corresponding to MC2a and MC2b. These results suggest that the vortex properties were different between MC1 and MC2.

As described above, using PAWR, we were able to capture the three-dimensional structure and changes in the vortices inside the convective cloud that caused the waterspouts. The continuous three-dimensional structure of the vortex is expected to make it possible to calculate the potential vorticity which helps to understand the formation process of such vortices. Currently, the case studies with this viewpoint are quite rare; further case studies are needed using PAWR.

**Author Contributions:** Conceptualization, R.I. and N.T.; methodology, R.I.; software, R.I. and N.T.; validation, R.I. and N.T.; writing—original draft preparation, R.I.; writing—review and editing, N.T.; visualization, R.I. and N.T. All authors have read and agreed to the published version of the manuscript.

**Funding:** This research received no external funding.

**Data Availability Statement:** PAWR data are available from National Institute of Information and Communications Technology (NICT) at https://pawr.nict.go.jp/index_en.html (accessed on 20 August 2022).

**Acknowledgments:** One of the authors (R.I.) thanks Tadayasu Ohigashi (National Research Institute for Earth Science and Disaster Prevention) who instructed him on how to handle KIN radar data. He also thanks Shinsuke Satoh (National Institute of Information and Communications Technology, NICT) who provided PAWR data processing programs. PAWR data was provided by the NICT. Information on waterspouts observed at the Naha Aviation Weather Station was provided by the Okinawa Meteorological Observatory.

**Conflicts of Interest:** The authors declare no conflict of interest.

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
