# Peer review of "Analysis of the Three-Dimensional Structure of the Misocyclones Generating Waterspouts Observed by Phased Array Weather Radar: Case Study on 15 May 2017 in Okinawa Prefecture, Japan"

_remotesensing, doi:10.3390/rs14215293_

Round 1
Reviewer 1 Report
This is a well presented paper on exciting high resolution vortex development.

Reviewer 2 Report
Recommendation: Minor revision
This is a well-written paper revealing the three-dimensional structure of the misocyclones generating waterspouts observed by phased array weather radar. The PAWR has been demonstrated to be capable of capturing detailed temporal changes in the internal structure of convective clouds that generate waterspouts. The general content of this paper is acceptable. I have some comments that may help to improve the physical understanding of this observational study.
Specific comments:
Line 12: (mesocyclone or mesocyclone, depending of their scale)
--> (mesocyclone or misocyclone, depending on their scale)
Line 183: Can the authors be more specific about how it is calculated? Does it mean a difference in the tangential velocity along the radial direction?
Line 189-195: PV is conserved in adiabatic and frictionless processes. Eq. 2 may be misreading as PV is defined as a constant. I suggest re-organizing this paragraph to separate the definition of PV and PV conservation law.
Line 198-200: This assumption claimed that vertical motion in a tornado is small. Is there any physical basis or reference?
Line 200-202: As I know, the PV expression as Eq. 3 is commonly derived using the shallow water approximation, which involves more assumptions in addition to the hydrostatic equilibrium assumption.
Line 209: How the vortex thickness h is retrieved or defined from the PAWR observation? Does the text in Lines 212-213 just describe the definition of thickness h?
Section 3.1: If some of the large-scale background figures were provided, it might be easier to follow these statements.
Line 388: Is there any reference demonstrating that such misocyclone can be viewed as adiabatic and frictionless? Otherwise, it is suspicious to claim the conservation law of PV here.
Conclusion and discussion: The advantages of PAWR in misocyclone detection are revealed: high temporal resolution and capability for PV estimation. Do these outcomes lead to any potential advance in the misocyclone formation or any conflict with previous observational or theoretical studies? It is beneficial to make a comparison with previous studies.
